# Molecular mechanism of voltage-dependent potentiation of KCNH potassium channels

**Gucan Dai, William N Zagotta\***

Department of Physiology and Biophysics, University of Washington, Seattle, United States

**Abstract** EAG-like (ELK) voltage-gated potassium channels are abundantly expressed in the brain. These channels exhibit a behavior called voltage-dependent potentiation (VDP), which appears to be a specialization to dampen the hyperexitability of neurons. VDP manifests as a potentiation of current amplitude, hyperpolarizing shift in voltage sensitivity, and slowing of deactivation in response to a depolarizing prepulse. Here we show that VDP of *D. rerio* ELK channels involves the structural interaction between the intracellular N-terminal eag domain and C-terminal CNBHD. Combining transition metal ion FRET, patch-clamp fluorometry, and incorporation of a fluorescent noncanonical amino acid, we show that there is a rearrangement in the eag domain-CNBHD interaction with the kinetics, voltage-dependence, and ATP-dependence of VDP. We propose that the activation of ELK channels involves a slow open-state dependent rearrangement of the direct interaction between the eag domain and CNBHD, which stabilizes the opening of the channel.

## Introduction

Ion channels in the KCNH family (EAG, ERG and ELK) are voltage-gated potassium channels important for nervous system function, cardiac physiology, and cancer biology (*Warmke and Ganetzky, 1994*; *Ganetzky et al., 1999*; *Pardo et al., 1999*; *Morais-Cabral and Robertson, 2015*) (*Figure 1—figure supplement 1*). ERG channels (Kv11) constitute the fast delayed rectifier in cardiomyocytes and are partly responsible for the repolarization of the cardiac action potential (*Sanguinetti et al., 1995*; *Trudeau et al., 1995*). EAG channels (Kv10) and ELK channels (Kv12) are abundantly and almost exclusively expressed in the brain where they also regulate electrical excitability, though their precise physiological function is not well understood (*Shi et al., 1998*; *Warmke and Ganetzky, 1994*; *Zou et al., 2003*; *Saganich et al., 2001*; *Martin et al., 2008*). Genetic deletion of ELK channels was shown to cause hippocampal hyperexcitability and epilepsy in mice (*Zhang et al., 2010*). In addition, EAG channels are also abundantly expressed in many forms of cancer (*Camacho, 2006*; *Pardo and StuhmerStühmer, 2014*).

Like other voltage-gated potassium channels, KCNH channels are composed of four subunits around a centrally located pore, where each subunit contains six transmembrane segments and an intracellular N-terminal and C-terminal region (*Figure 1A*). Although the KCNH channels contain a cyclic nucleotide-binding homology domain (CNBHD) in the C-terminal region, the channels do not bind and are not directly regulated by cyclic nucleotides, including cAMP and cGMP (*Brelidze et al., 2009*; *Robertson et al., 1996*). Instead, the analogous cyclic nucleotide-binding pocket of the CNBHD is occupied by an 'intrinsic ligand' from a short sequence at the C-terminal end of the CNBHD (*Marques-Carvalho et al., 2012*; *Brelidze et al., 2012*). This intrinsic ligand regulates KCNH channel function (*Marques-Carvalho et al., 2012*; *Brelidze et al., 2012*; *Zhao et al., 2017*)

**\*For correspondence:** zagotta@uw.edu

**Competing interests:** The authors declare that no competing interests exist.

**eLife digest** In humans and other animals, electrical signals trigger the heart to beat and carry information around the brain and nervous system. Particular cells can generate these signals by regulating the flow of ions into and out of the cell via proteins called ion channels. These proteins sit in the membrane that surrounds the cell and will open or close in response to specific signals. For example, an ion channel in humans called hERG allows positively-charged potassium ions to flow out of a heart cell to help the cell return to its "resting" state after producing an electrical signal. Defects in hERG can alter the rhythm at which the heart beats, leading to a serious condition called Long QT syndrome.

The human hERG channel is part of a family of related channels known as the KCNH channels. These channels are made of four protein subunits that assemble to form a pore that spans the cell membrane. When a cell is resting before producing an electrical signal, KCNH channels are generally closed. However, once an electrical signal starts, the flow of ions through other ion channels in the cell membrane changes an electrical property across the membrane known as the "voltage". This change in voltage causes KCNH channels to open.

Dai and Zagotta studied how a KCNH channel known as ELK from zebrafish responds to changes in membrane voltage. The experiments show that the manner in which ELK channels respond to the voltage is due to changes in how the subunits interact in the part of the channel that lies inside the cell. Further experiments using several new techniques reveal in much more detail how the shape of the channel alters as the voltage changes. These new techniques could also be used to observe how other KCNH channels in the heart and brain change shape in response to changes in voltage. This could lead to the design of new drugs to treat heart and neurological diseases.

and explains, in part, why KCNH channels are not regulated by cyclic nucleotides. Another important structural feature of KCNH channels is the interaction between the N-terminal eag domain (PAS domain and PAS cap) and C-terminal CNBHD (*Gianulis et al., 2013*; *Gustina and Trudeau, 2009*; *Haitin et al., 2013*; *Whicher and MacKinnon, 2016*) (*Figure 1A*). This interaction has been demonstrated to be critical for the proper function of KCNH channels. Mutations in KCNH channels that impair this eag domain-CNBHD interaction lead to alterations in channel trafficking and gating, which are thought to underlie some forms of Long QT Syndrome and cancer (*Curran et al., 1995*; *Gustina and Trudeau, 2009*).

One behavior shared by ERG and ELK channels is mode shift or hysteresis (*Li et al., 2015*; *Tan et al., 2012*; *Goodchild et al., 2015*). This electrical property is characterized by a shift in the voltage dependence of activation to more hyperpolarized voltages in response to a depolarizing prepulse. In ERG channels, this mode shift is thought to be responsible for the slowing of deactivation that contributes to the repolarization of the cardiac action potential (*Sanguinetti et al., 1995*; *Trudeau et al., 1995*). This phenomenon, also called prepulse facilitation, has been found in other types of ion channels including N-type and P/Q-type calcium channels and HCN channels (*Hoshi et al., 1984*; *Hoshi and Smith, 1987*; *Bean, 1989*; *Männikkö et al., 2005*; *Elinder et al., 2006*). We refer to this behavior in KCNH channels as voltage-dependent potentiation (VDP).

In this paper, we studied the structural mechanism underlying the VDP in ELK channels. Using deletions, mutations, and chimeric constructs we show that VDP involves the interaction between the eag domain and CNBHD. To measure the distance between positions in the eag domain and CNBHD, we used transition metal ion FRET (tmFRET) (*Taraska et al., 2009a*) together with incorporation of a fluorescent noncanonical amino acid (*Chatterjee et al., 2013*). By simultaneously measuring channel current and tmFRET using patch-clamp fluorometry (PCF) (*Zheng and Zagotta, 2003*), we showed that the distance between the eag domain and CNBHD decreases with the time course, voltage-dependence, and ATP-dependence of VDP. These results indicate that VDP in ELK channels involves a slow rearrangement of the interaction between the eag domain and the CNBHD that is coupled to channel opening.

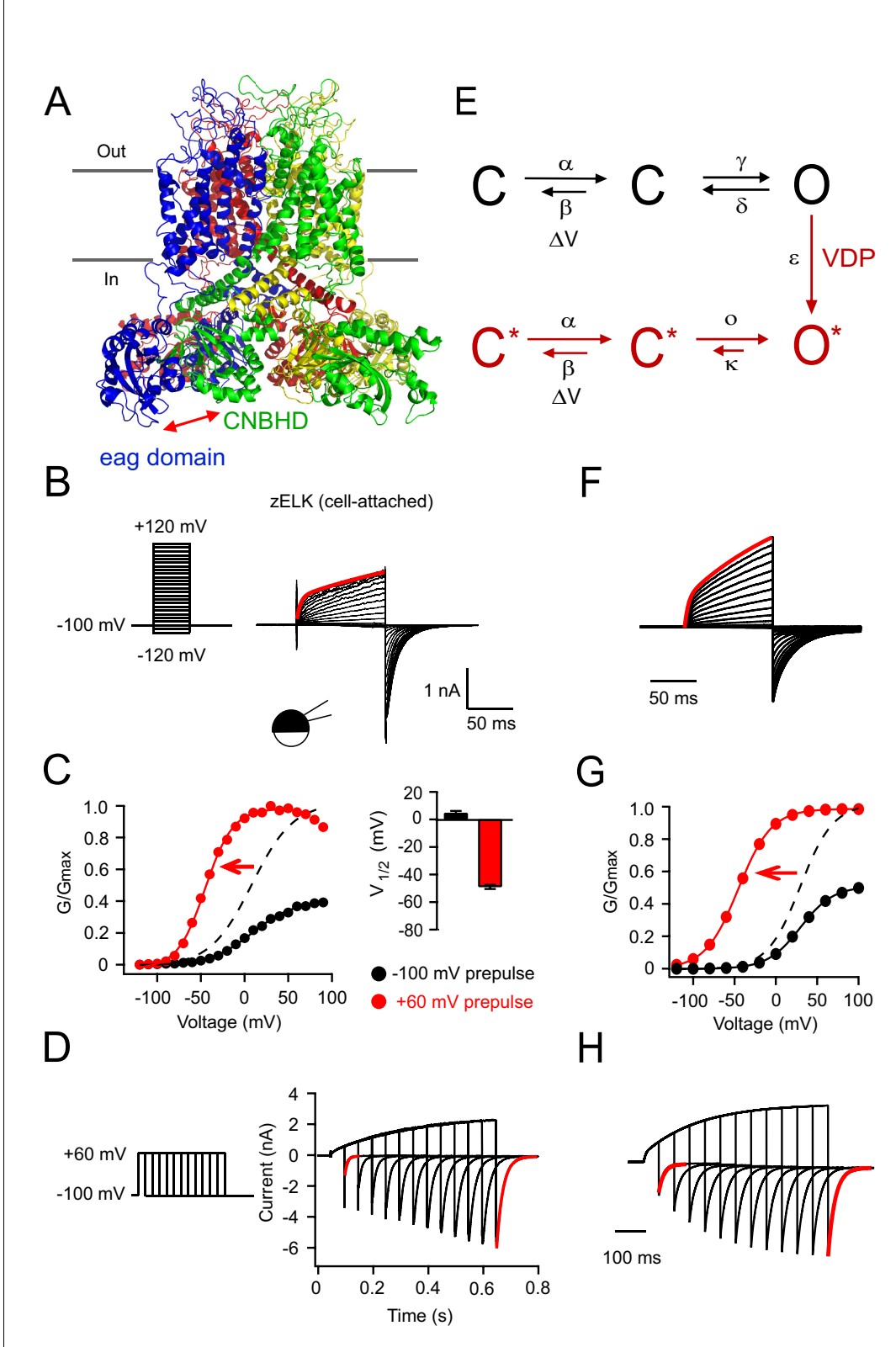

**Figure 1.** VDP of zELK channels. (**A**) Homology model of the structure of the zELK channel illustrating the intersubunit eag domain-CNBHD interaction (side-view parallel to the plasma membrane), based on the cryo-EM structure of the rEAG1 channel (PDB code: 5K7L) (*Whicher and MacKinnon, 2016*). Red arrow highlights the direct eag domain-CNBHD interaction. (**B**) Representative current-voltage (I–V) recordings of zELK channels in the cell-attached configuration using the voltage protocol on the left. The red trace is the double-exponential fitting of the current elicited by a +120 mV

*Figure 1 continued on next page*

*Figure 1 continued*

voltage pulse ($\tau 1$ = 4 ms and $\tau 2$ = 206 ms). (C) Representative conductance-voltage (G–V) curves of zELK channels in the cell-attached configuration without and with a +60 mV prepulse. The dashed curve is the same data as the black solid curve but normalized to the amplitude of the red curve. (Right) Summary of the $V_{1/2}$ of the G–V curves from multiple patches (n = 4–14). (D) zELK currents (right) elicited by a voltage protocol with increasing durations of +60 mV pulse (left). The deactivation time constants for the red traces are 8.5 and 22.9 ms respectively). (E) A 6-state kinetic model for the VDP of KCNH channels. (F–H) Simulated data based on the 6-state model for zELK channels using QuB software (State University of New York at Buffalo) using the same protocols as panels B–D respectively. Voltage-dependent rate constants are given by k(V) = k0 exp(k1V), where V is voltage, k0 is the rate at 0 mV, and k1 the voltage dependence of the rate. For the forward rate constant $\alpha$ of the voltage-dependent transition: k0 = 80 s$^{-1}$ and k1 = 0.025 mV$^{-1}$; for the reverse rate constant $\beta$ of the voltage-dependent transition: k0 = 600 s$^{-1}$ and k1 = $-$0.025 mV$^{-1}$. For the rate constant of the VDP transition step: $\varepsilon$ = 35 s$^{-1}$. For the other transitions illustrated: $\gamma$ = 60 s$^{-1}$, d = 200 s$^{-1}$, $o$ = 5000 s$^{-1}$, $\kappa$ = 70 s$^{-1}$.

The following figure supplement is available for figure 1:

**Figure supplement 1.** Dendrogram of KCNH channel family.

## Results

### VDP of zELK channels

For this study, we used a vertebrate ELK channel from zebrafish (zELK) which exhibits robust expression in heterologous expression systems (*Figure 1—figure supplement 1*). Previously, the basic electrophysiological properties of zELK channels were shown to be similar to the mammalian orthologs, and the structure of the C-linker/CNBD of the channel was solved by X-ray diffraction (*Brelidze et al., 2012*). zELK channels were expressed in Xenopus oocytes and activated by depolarizing voltage steps from $-120$ mV to $+120$ mV in the cell-attached patch-clamp configuration (*Figure 1B*). As for other KCNH channels, zELK is a K$^+$-selective channel activated by membrane depolarization with prominent inward tail currents seen with high concentrations of potassium in the recording electrode.

VDP of zELK channels manifests in three ways. The first is that the activation of zELK channels at depolarizing voltages exhibits prominent double exponential kinetics, with a fast (~4 ms) and a slow (~200 ms) component (*Figure 1B*). This suggests that prolonged depolarization is causing the channel to transition to a second more stable open conformation. The second manifestation of VDP is hysteresis in the steady-state conductance-voltage (G-V) curve. We applied a 500 ms depolarizing prepulse to +60 mV before a family of voltage steps. The depolarizing prepulse caused about a $-60$ mV shift in the G-V curve to more hyperpolarized voltages as well as a dramatic increase in the peak tail-current amplitude (*Figure 1C*). The third manifestation of VDP is apparent by varying the duration of depolarizing voltages (+60 mV) and monitoring the amplitude and time course of the tail current at $-100$ mV. The tail current amplitude increased $3.5 \pm 0.3$ fold (n = 11) for depolarizing voltage pulses of longer durations, with a time course that closely matched the slow component of the activation kinetics (*Figure 1D*). More interestingly, the time constant of the tail current increased from $8.4 \pm 0.4$ ms to $16.9 \pm 1.5$ ms with longer pulses (n = 12) (*Figure 1D*). This suggests that channels closed more slowly from a potentiated open state.

These three manifestations of VDP can be recapitulated in a simple kinetic scheme (*Figure 1E*). In this scheme the voltage-dependent activation of the channel was modeled as a single voltage-dependent transition followed by a voltage-independent closed-to-open transition. VDP was then modeled as a slow transition to a potentiated mode with an unaltered voltage-dependent transition but a more favorable closed-to-open transition. As required by thermodynamics, the mode shift is more favorable from the open state than from the closed states, and is therefore coupled to activation. This coupling produces a voltage-dependence to the mode shift, and therefore VDP. While clearly oversimplified, this simple gating scheme could quantitatively account for the double exponential activation (*Figure 1F*), the shift in the G-V curve with depolarizing prepulses (*Figure 1G*), and the slowing of the tail currents with longer depolarizing pulses (*Figure 1H*). In the rest of the paper, we determine the molecular mechanism that underlies the mode shift that produces VDP.

## VDP is eliminated in the inside-out patch-clamp configuration

zELK channels exhibit a dramatic run-up in activity after patch excision. Similar to VDP, this run-up manifested as an increase in tail current amplitude (a $2.2 \pm 0.3$ fold increase compared to the cell-attached configuration, measured after +120 mV depolarization, n = 7) (*Figure 2A*) and a significant shift of the G-V curve of channel activation to more hyperpolarized voltages ($V_{1/2} = -52.5 \pm 3.5$ mV, n = 6) (*Figure 2C*). The shift of the G-V curve happened gradually after excision and reached steady-state after about 20 mins (*Figure 2B*). Interestingly, the VDP was almost completely eliminated in excised patches, with no further shift in the G-V curve with depolarizing prepulses (*Figure 2C*), and no slowing of the tail currents with longer depolarizing pulses (*Figure 2D,E*). This suggests that patch excision shifted the channels into the potentiated mode even without a depolarizing prepulse. Patch excision also revealed a prominent voltage-dependent inactivation in zELK channels, particularly at very depolarized voltages (>+60 mV) (*Figure 2A*).

Patch cramming, i.e. inserting the excised patch back into the intracellular regions of the oocyte, was able to completely restore the cell-attached channel behavior within five mins, indicating some cytosolic factors were lost in the inside-out configuration (*Figure 2A and B*). Reducing reagent DTT in the bath did not prevent the run-up, suggesting disulfide bonding was not involved in this run-up (*Figure 2—figure supplement 1A*). Previously, it was proposed that run-up of human ELK1 channels was mediated by $PI(4,5)P_2$ hydrolysis (*Li et al., 2015*). We found that supplementation of the bath solution with 2 mM $ATP/Mg^{2+}$ was able to prevent the run-up and maintain the VDP observed in the cell-attached configuration (*Figure 2F,G and H*). ATP alone without $Mg^{2+}$ was not able to prevent the run-up (*Figure 2—figure supplement 1B*). These results suggest that, consistent with previous findings in human ELK (*Li et al., 2015*), $PI(4,5)P_2$ is also required for VDP in zELK, and hydrolysis of $PI(4,5)P_2$ after patch excision leaves the channel in a potentiated state.

## Interaction between the eag domain and CNBHD controls the VDP of zELK channels

To determine the role of the intracellular eag domain and CNBHD in VDP, we made mutations in these domains and tested for VDP. We found the VDP was almost completely abolished in zELK channels with the eag domain deleted (zELK $\Delta$eag) (*Figure 3A*). In the cell-attached configuration; the average change in $V_{1/2}$ ($\Delta V_{1/2}$) with a +60 mV prepulse was $-12.7 \pm 1.4$ mV (n = 7) (*Figure 3B*) compared to $-54.0 \pm 2.0$ mV (n = 14) in wild-type zELK. In addition, zELK $\Delta$eag did not show an increase in the time constant of deactivation with longer depolarizing pulses (*Figure 3C*) as seen in the wild-type channel.

We next mutated an intersubunit salt bridge between the eag domain and the CNBHD predicted based on structures of the EAG1 channel (*Figure 3D*) (*Haitin et al., 2013*; *Whicher and MacKinnon, 2016*). In zELK, the homologous positions of the salt-bridging residues are R57 in the eag domain and D681 in the CNBHD. We found that charge reversal mutations (zELK-R57D or D681R) that would disrupt the salt bridge not only shifted the initial $V_{1/2}$, but also attenuated the VDP (*Figure 3E and F*). When we made the charge-swapping mutations (zELK-R57D, D681R) by combining the individual reversal mutations, we partially rescued the channel behavior to that of the wild-type channels (*Figure 3E and F*). These results indicate that the eag domain and CNBHD are interacting via the salt bridge and this interaction is supporting the VDP.

To test the hypothesis that the intracellular eag domain-CNBHD interaction is sufficient to confer VDP to the channel, we engineered a chimeric channel with the S1 to S6 transmembrane domains of mEAG1 (amino acids: 209–503) and N- (amino acids: 1–217) and C- (amino acids: 544–914) terminal regions from zELK. Wild-type mouse EAG1 channels do not have VDP (*Figure 3G,H*). However, the mEAG1-zELK chimera exhibited prominent VDP (*Figure 3I*). The average $\Delta V_{1/2}$ with the +60 mV prepulse was $-39.1 \pm 2.6$ mV (n = 9) for the mEAG1-zELK chimera in the cell-attached configuration. These results suggest that the eag domain-CNBHD complex of zELK is sufficient to confer VDP on mEAG1.

## Measuring the distance between eag domain and CNBHD using tmFRET

To determine if there is a rearrangement of the eag domain-CNBHD interaction during VDP, we used transition metal ion FRET (tmFRET) combined with patch-clamp fluorometry. tmFRET measures

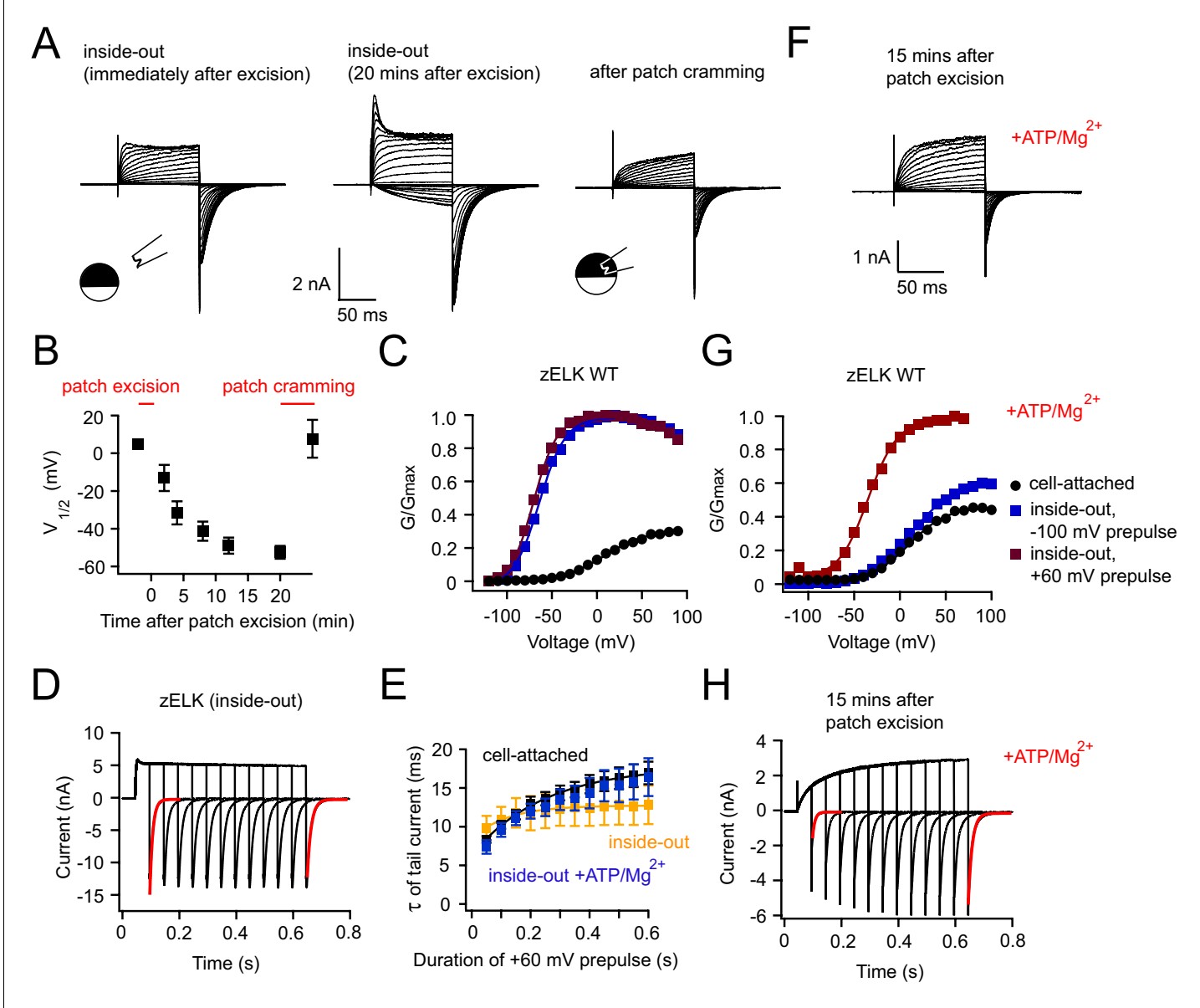

**Figure 2.** Run-up of zELK channels and loss of VDP after patch excision. (**A**) Representative I-V recordings of zELK channels immediately after excision (left) and 20 mins after excision (middle) in the inside-out configuration, as well as after patch cramming (right) using the same voltage protocol illustrated in *Figure 1B*. (**B**) Time course of the $V_{1/2}$ change of the G-V curve of zELK channels after patch excision; patch-cramming restored the $V_{1/2}$ to that before patch excision (n = 3–6). (**C**) Representative G-V curves of zELK channels in the cell-attached configuration (black), in the inside-out configuration with a −100 mV prepulse (blue), and with a +60 mV prepulse (red) (see the legend in panel **G**). (**D**) zELK currents elicited by a voltage protocol with increasing durations of +60 mV pulse in the inside-out configuration (same protocol illustrated in *Figure 1D*). The deactivation time constants for the red traces are 10.3 and 13.7 ms respectively). (**E**) Plot of the time constants of deactivation versus the duration of the +60 mV pulse for zELK channels in inside-out patches with and without ATP/Mg$^{2+}$ (n = 4). The corresponding data for the cell-attached configuration is shown for comparison. (**F**) I-V recordings of zELK channels in inside-out patches with ATP/Mg$^{2+}$ in the bath solution showing no run-up after patch excision. (**G**) G-V curves of zELK channels in the same conditions as panel **C** with the addition of 2 mM ATP/Mg$^{2+}$ to the bath solution. (**H**) zELK current elicited using the same voltage protocol in *Figure 1D*, with 2 mM ATP/Mg$^{2+}$ in the bath solution. The deactivation time constants for the red traces are 7.2 and 14.3 ms, respectively.

The following figure supplement is available for figure 2:

**Figure supplement 1.** Run-up of zELK channels is not prevented by reducing agents or ATP without Mg$^{2+}$.

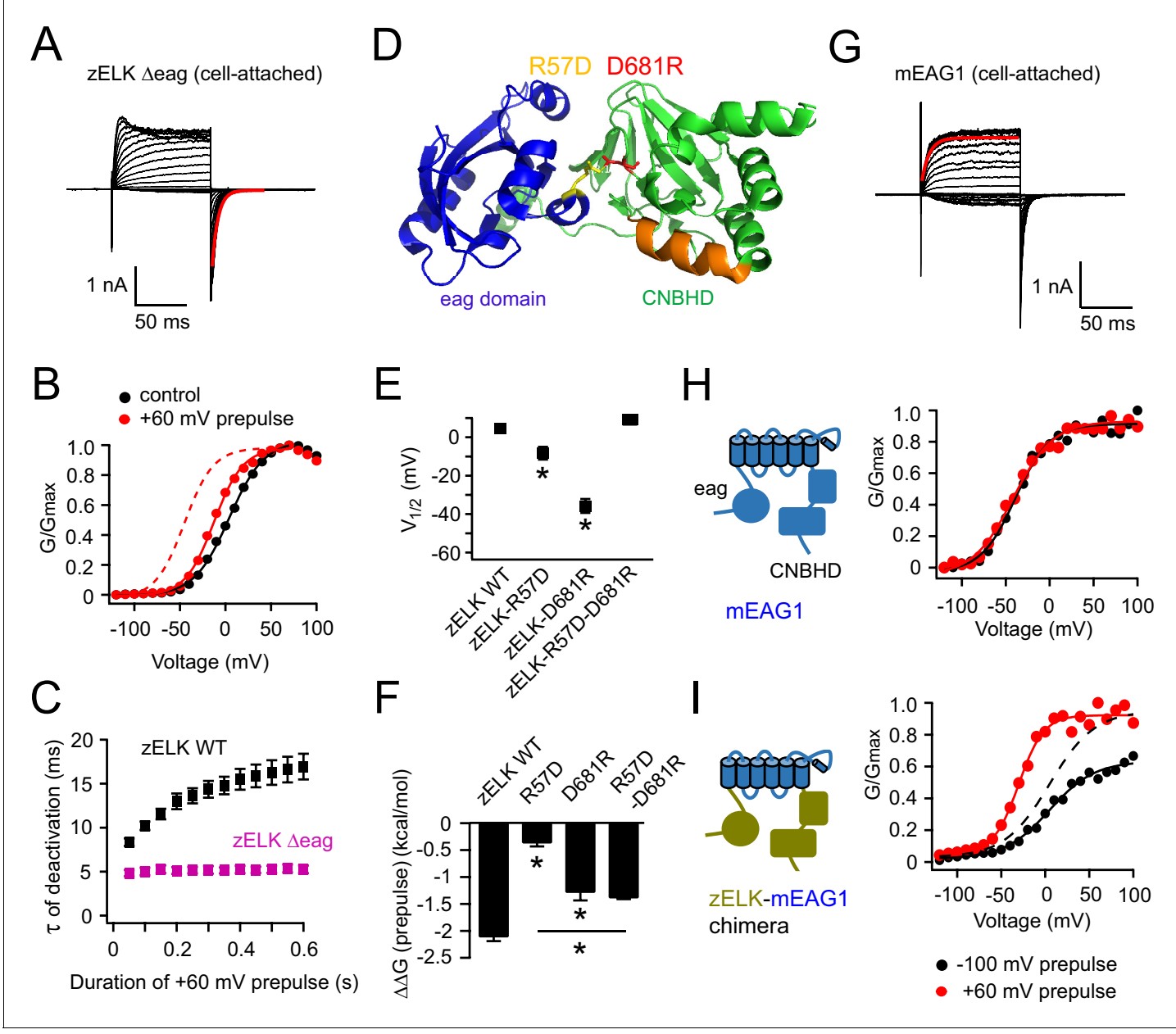

**Figure 3.** Structural perturbations of the eag domain and CNBHD impair VDP. (**A**) I-V recordings of zELK Δeag channels in the cell-attached configuration (τ = 5.9 ms for the faster deactivation highlighted in red). (**B**) Representative G-V curves for zELK Δeag channels without or with a +60 mV prepulse. The dashed trace illustrates the G-V curve of the wild-type channel after a +60 mV prepulse. (**C**) Plot of the time constant for deactivation vs. the duration of the +60 mV pulse for zELK Δeag channels. (**D**) Ribbon structure of the eag domain/CNBHD complex of mEAG1 channels (PDB code: 4LLO) (**Haitin et al., 2013**), highlighting a salt bridge between the eag domain and CNBHD formed by R57 and D681 in the analogous positions of zELK. (**E**) and (**F**) Summary of effects of salt-bridge mutations on $V_{1/2}$ (**E**) and VDP measured by $\Delta\Delta G$ (prepulse) (**F**), (n = 4–5). *p<0.05. (**G**) Representative I-V recordings of mEAG1 channels showing that the kinetics of activation has only one component (τ = 7.7 ms for the red trace). The fit is applied to the exponential activation following small sigmoidal delay. (**H**) G-V curves of mEAG1 channels in the same conditions as panel B. (**I**) G-V curves of a zELK-mEAG1 chimera containing the N- and C-terminal intracellular domains from zELK and transmembrane (S1–S6) domain from mEAG1. The dashed curve is the same data as the black solid curve but normalized to the amplitude of the red curve.

the FRET between a donor fluorophore and a nonfluorescent transition metal ion acceptor (*Latt et al., 1972*; *Horrocks et al., 1975*; *Richmond et al., 2000*; *Sandtner et al., 2007*; *Taraska et al., 2009a*, *2009b*). The efficiency of tmFRET is steeply dependent on the distance between the donor fluorophore and the acceptor metal ion and can be directly measured from the percent quenching of the donor's fluorescence upon addition of the metal ion. Compared to traditional FRET, tmFRET measures much shorter distances (10–20 Å) and has less orientation dependence, making it ideal for measuring intramolecular distances in proteins (*Taraska et al., 2009a*).

As the donor fluorophore for tmFRET, we used the fluorescent noncanonical amino acid L-Anap (*Figure 4A*). L-Anap was site-specifically incorporated into zELK channels using the *amber* (TAG) stop-codon suppression strategy (*Chatterjee et al., 2013*; *Kalstrup and Blunck, 2013*; *Aman et al., 2016*; *Sakata et al., 2016*; *Zagotta et al., 2016*). As the tmFRET acceptor, we used $Co^{2+}$ coordinated by a dihistidine pair engineered into an α helix in zELK (*Figure 4A*). The emission spectrum of L-Anap overlaps with the absorption spectrum of $Co^{2+}$-dihistidine, predicting a distance for 50% FRET efficiency ($R_0$) of 12 Å (*Figure 5—figure supplement 1*) (*Zagotta et al., 2016*; *Aman et al., 2016*).

When combined with patch-clamp fluorometry (*Zheng and Zagotta, 2003*), tmFRET is able to detect the distance change between protein domains with simultaneous electrophysiological measurements while controlling the membrane voltage and intracellular solution. L-Anap was incorporated into the zELK eag domain by mutating the codon for amino acid 51 in the A helix to the amber stop codon TAG (*Figure 4A,B*). We also fused a YFP at the C-terminal end of the zELK channels as a fluorescent reporter to confirm the successful expression of the full-length channel (*Figure 4B*). Xenopus oocytes were then injected with the zELK-E51TAG-YFP mRNA, L-Anap, and a plasmid pANAP coding for the orthogonal *amber* suppressor tRNA/aminoacyl-tRNA synthetase (aaRS) pair for L-Anap (*Chatterjee et al., 2013*; *Kalstrup and Blunck, 2013*; *Aman et al., 2016*; *Sakata et al., 2016*; *Zagotta et al., 2016*). Only patches from oocytes injected with all three components exhibited Anap fluorescence (*Figure 4C*). The linear correlation between the Anap fluorescence and the YFP fluorescence and current in the patches indicates that virtually all of the Anap fluorescence was coming from L-Anap incorporated within the functional channel and not from any nonspecific background fluorescence (*Figure 4D and E*). Indeed, the negative controls in the absence of channels or with wild-type zELK channels (no TAG mutation) produced negligible Anap fluorescence (*Figure 4C*).

With E51Anap located within the A helix of the eag domain, a dihistidine (K729H, E733H) was introduced to the C helix of the CNBHD (*Figure 4A and B*). The tmFRET efficiency between L-Anap and $Co^{2+}$ bound to the dihistidine was measured by the degree of quenching of Anap fluorescence by $Co^{2+}$. With an increasing concentration of $Co^{2+}$ applied to the intracellular face of excised patches held at −100 mV, the Anap fluorescence decreased monotonically (*Figure 5A*). The $Co^{2+}$-mediated quenching was reversed by applying 10 mM EDTA to chelate the divalent cations (*Figure 5—figure supplement 2*). This quenching of Anap fluorescence by $Co^{2+}$ is indicative of FRET between the L-Anap and the bound $Co^{2+}$. The channel without the dihistidine produced only a small amount of quenching up to 1 mM $Co^{2+}$ (*Figure 5A*) and was used to correct the FRET efficiency for any quenching that was not due to $Co^{2+}$ binding to the dihistidine (see tmFRET efficiency calculation in the Materials and methods) (*Figure 5B*). The apparent tmFRET efficiency increased with increasing $Co^{2+}$ concentration and was well described by a Langmuir isotherm with an apparent affinity of around 70 μM, and a maximal FRET efficiency of 0.71 ± 0.05 (n = 4). These results suggest that $Co^{2+}$ binding to the dihistidine in the CNBHD is in close proximity to L-Anap in the eag domain, as predicted from the X-ray crystal and cryoEM structures (*Haitin et al., 2013*; *Whicher and MacKinnon, 2016*).

## Rearrangements between the eag domain and CNBHD have the ATP/$Mg^{2+}$-dependence, voltage-dependence, and time course of VDP

We showed above that zELK channels in patches excised in the standard saline solution are potentiated and lose VDP, while channels excised in the presence of ATP/$Mg^{2+}$ maintain VDP (*Figure 2*). zELK-E51Anap, K729H-E733H channels behaved similarly to wild-type zELK channels and exhibited VDP in ATP/$Mg^{2+}$ (*Figure 5C*). In the presence of ATP/$Mg^{2+}$, zELK-E51, K729H-E733H channels exhibited a shift in the G-V curve ($\Delta V_{1/2}$) of −43.5 ± 2.7 mV with depolarizing prepulses, only slightly less than the wild-type channel in the cell-attached configuration (*Figure 5D*). We next measured

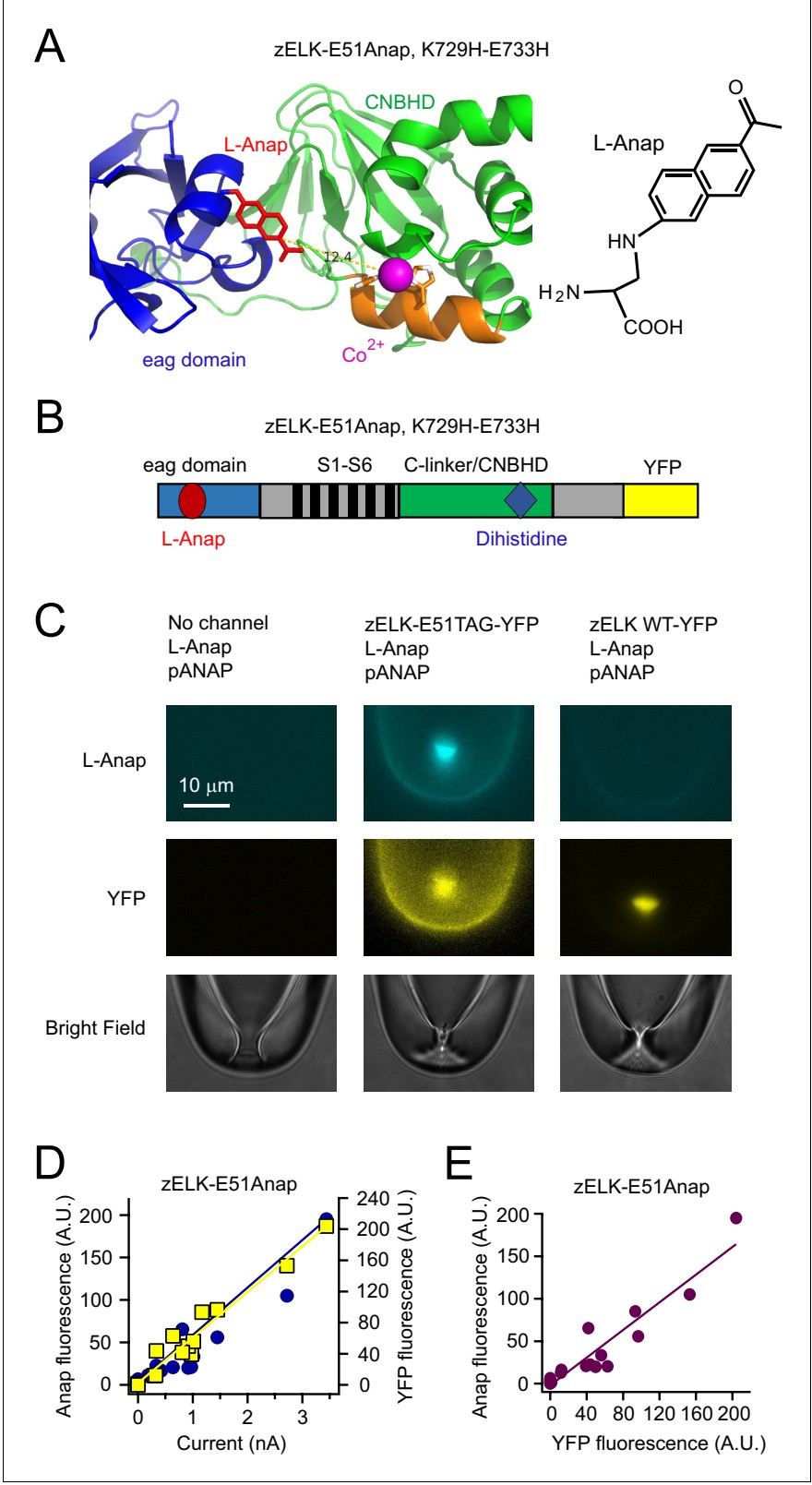

**Figure 4.** Strategy of combining tmFRET, patch-clamp fluorometry and a fluorescent noncanonical amino acid L-Anap to study conformational changes of zELK channels. (**A**) Ribbon diagram of eag domain-CNBHD complex illustrating the strategy of using tmFRET between an noncanonical amino acid L-Anap (structure shown on the right) and Co²⁺ chelated by dihistidines to measure interdomain (intramolecular) distances. (**B**) Cartoon illustrating

*Figure 4 continued on next page*

*Figure 4 continued*

the zELK channel construct with L-Anap site, dihistidine site, and C-terminal YFP. (C) Representative patch-clamp fluorometry images showing the specific incorporation of L-Anap into zELK channels using the amber stop-codon suppression strategy. (D) and (E) L-Anap fluorescence in patches correlated with zELK channel current (D) or YFP fluorescence (E).

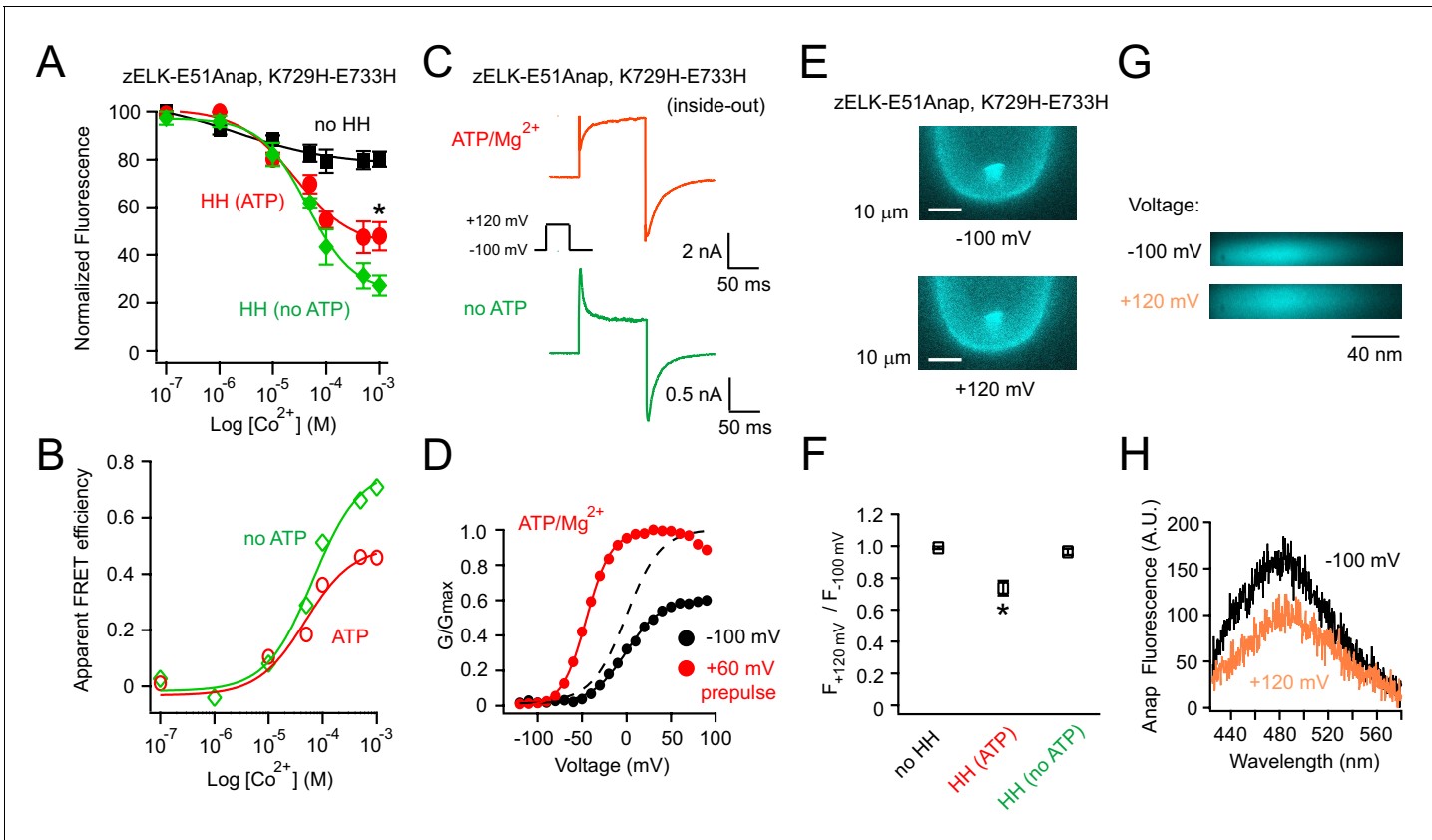

**Figure 5.** Measuring the ATP/Mg$^{2+}$-dependent and voltage-dependent change in the distance between the eag domain and CNBHD of zELK channels using tmFRET. (A) Quenching of Anap fluorescence measured using PCF by different concentrations of Co$^{2+}$ with or without dihistidines and in the absence and presence of 2 mM ATP/Mg$^{2+}$ at the resting holding voltage of −100 mV. (B) tmFRET efficiency as a function of Co$^{2+}$ concentration in the absence and presence of ATP/Mg$^{2+}$ as described. The smooth curves are fits of the Langmuir isotherm, Apparent FRET$_{eff.}$ = FRET$_{eff}$ [Co$^{2+}$] / (K$_{1/2}$ + [Co$^{2+}$]), with the following parameters: FRET$_{eff}$ = 0.46, K$_{1/2}$ = 48.0 μM with ATP (red) and FRET$_{eff}$ = 0.71, K$_{1/2}$ = 66.1 μM without ATP (green). For the control construct zELK-E51Anap without the dihistidine, the quenching data with and without ATP/Mg$^{2+}$ were merged since ATP/Mg$^{2+}$ did not produce any significant difference. (C) Inside-out patch-clamp recordings of zELK E51Anap, K729H-E733H channels exhibiting a ATP/Mg$^{2+}$-dependent slow component of activation typical of VDP in wild-type zELK channels. (D) Representative G-V curves of zELK-E51Anap, K729H-E733H channels exhibiting prepulse-dependent shift in the voltage-dependence of activation typical of VDP in wild-type zELK channels (different patch from panel C). (E) Representative PCF images showing Anap fluorescence decreased when the membrane voltage was stepped from −100 mV to +120 mV for zELK-E51Anap, K729H-E733H channels in the presence of 1 mM Co$^{2+}$ and ATP/Mg$^{2+}$ in the bath. (F) Summary data showing the Anap fluorescence decrease by depolarization was abolished without the dihistidines or when the VDP disappeared in the absence of ATP/Mg$^{2+}$ (n = 5), *p<0.05. The fluorescence was measured using a bandpass filter for Anap emission. (G) Spectral images of L-Anap emission at −100 mV and +120 mV. (H) Emission spectra from the spectral images shown in panel G.

The following figure supplements are available for figure 5:

**Figure supplement 1.** tmFRET between L-Anap and transition metal ions.

**Figure supplement 2.** Reverse of Co$^{2+}$ quenching by EDTA.

the tmFRET efficiency between sites in the eag domain and CNBHD in the absence and presence of ATP/Mg$^{2+}$ (*Figure 5A and B*). FRET efficiency was determined at a saturating concentration of 1 mM Co$^{2+}$ where the dihistidine sites are expected to be completely bound by Co$^{2+}$ (see tmFRET efficiency calculation in the Materials and methods). We found that, with ATP/Mg$^{2+}$ added to the bath, the FRET efficiency at −100 mV decreased to $0.46 \pm 0.08$ (n = 4) compared to $0.71 \pm 0.05$ without 2 mM ATP/Mg$^{2+}$ added (*Figure 5B*). Using the Förster equation and an R$_0$ of 12 Å, this corresponds to a distance change of 2.0 Å. These results suggest that, at hyperpolarizing voltages, these two sites within the eag domain and CNBHD are further apart when the channel is not potentiated and closer together when the channel is potentiated.

We next measured the voltage-dependence of the conformational change between the eag domain and the CNBHD. With patch-clamp fluorometry, we found the steady-state Anap fluorescence intensity with 1 mM Co$^{2+}$ decreased at +120 mV compared to −100 mV (*Figure 5E and F*). This decrease was not present without dihistidine or in the absence of ATP/Mg$^{2+}$ (*Figure 5F*). L-Anap is an environmentally sensitive fluorophore whose emission spectrum shifts to shorter wavelengths in more hydrophobic environments (*Chatterjee et al., 2013*). To determine if the decreased fluorescence was associated with a change of the environment of L-Anap, we measured the emission spectra of Anap fluorescence in patches at −100 mV and +120 mV. We found the wavelength of the L-Anap peak emission was not significantly shifted despite the reduction in the intensity of peak emission produced by the +120 mV voltage pulse in the presence of 1 mM Co$^{2+}$ (*Figure 5G and H*). Together with the absence of a fluorescence change without a dihistidine (*Figure 5F*), these results are consistent with a FRET mechanism for Co$^{2+}$ quenching and not a change in environment (*Figure 5G and H*). These results indicate that, similar to VDP, membrane depolarization causes a ATP/Mg$^{2+}$-dependent and voltage-dependent rearrangement between the eag domain and CNBHD. This suggests that the rearrangement between the eag domain and the CNBHD is associated with VDP.

To further test that the rearrangement between the eag domain and CNBHD is associated with VDP, we measured the kinetics of the change in tmFRET and compared it with kinetics of the development and the recovery of VDP. Using patch-clamp fluorometry, we simultaneously measured the time course of the development of VDP and the time course of the domain rearrangement in zELK-E51, K729H-E733H channels. Fluorescent images were captured every 100 ms with a 50 ms exposure time. In the presence of ATP/Mg$^{2+}$ and 1 mM Co$^{2+}$, the Anap fluorescence decreased with a +60 mV depolarization and recovered after repolarization to −100 mV (*Figure 6A*). The time constant, $261 \pm 56$ ms, for the decrease in Anap fluorescence was not statistically different from the time constant of approximately $308 \pm 26$ ms for the slow component of channel activation associated with the development of VDP (*Figure 6A and B*, also *Figure 1D*). In the absence of ATP/Mg$^{2+}$, the channel activated quickly with the +60 mV depolarization without a slow component, and the concurrent Anap fluorescence was unchanged by the voltage step (*Figure 6A*).

The time course of the recovery of tmFRET also closely matched the time course of the recovery of VDP. The time constant for recovery of the Anap fluorescence at −100 mV was $137.5 \pm 16$ ms (*Figure 6A and C*). To measure the recovery rate of VDP, we used a new voltage protocol applying a −100 mV recovery pulse of variable durations after a +60 mV prepulse (*Figure 6—figure supplement 1A*). With an increased duration of the −100 mV pulse, the VDP gradually disappeared; a 500 ms recovery pulse shifted the V$_{1/2}$ back to the control value without the +60 mV prepulse (*Figure 6—figure supplement 1*). The recovery of the peak tail-current amplitude happened in a similar but somewhat slower time frame compared to the V$_{1/2}$ recovery (*Figure 6—figure supplement 1B and C*). The time constant of the V$_{1/2}$ recovery of wild-type channels was not significantly different from the time constant of the recovery of Anap fluorescence ($p > 0.05$, *Figure 6C*). Overall these tmFRET experiments demonstrate that there is a rearrangement between the eag domain and CNBHD that exhibits the same ATP/Mg$^{2+}$-dependence, voltage-dependence, and kinetics as VDP. Combined with our finding that VDP is altered or eliminated by mutations of the eag domain and CNBHD (*Figure 3*), these experiments suggest that the VDP is produced partially or fully by a slow open-state dependent rearrangement of the direct interaction between the eag domain and CNBHD, which stabilizes the opening of the channel.

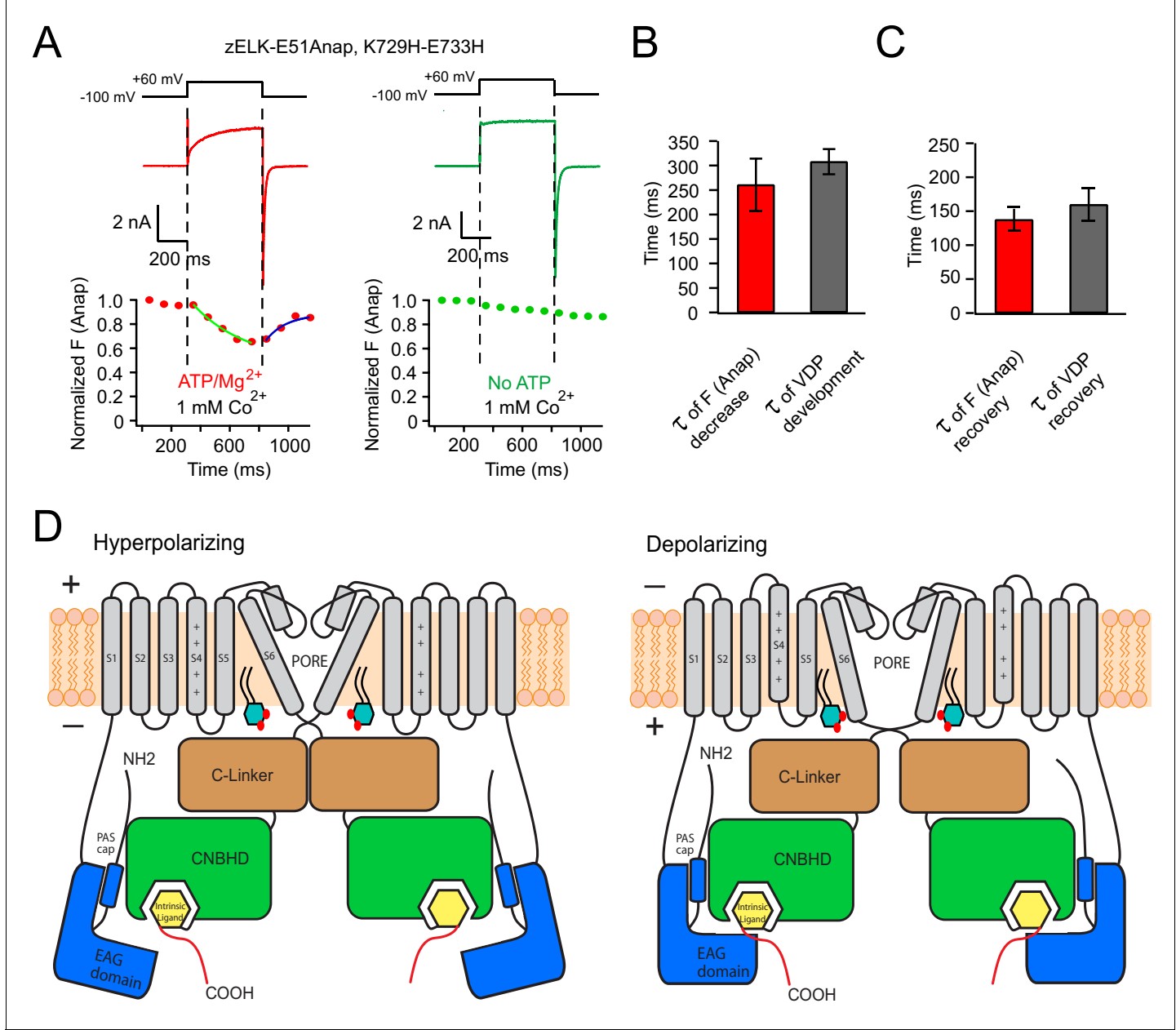

**Figure 6.** Voltage-dependent rearrangement of the eag domain-CNBHD interaction for zELK channels. (**A**) Kinetic measurement of Anap fluorescence during a +60 mV depolarization pulse with simultaneous current recordings, in the presence or absence of ATP/Mg$^{2+}$. (**B**) and (**C**) Comparison of the kinetics of the development (**B**) and the recovery (**C**) of tmFRET and VDP (n = 4–6). (**D**) Cartoon illustrating that the VDP of zELK channels involves a rearrangement of the direct interaction between the eag domain and CNBHD. The intersubunit interaction of the eag domain and CNBHD of diagonal subunits is illustrated at hyperpolarizing and depolarizing voltages showing a rearrangement of this interaction accompanies the VDP. Other changes such as pore opening and movement of the S4 are also important for channel activation. The yellow box indicates the intrinsic ligand. PI(4,5)P$_2$ in the inner leaflet of the plasma membrane is shown to highlight its potential role in regulating VDP.

The following figure supplement is available for figure 6:

**Figure supplement 1.** Measuring the recovery of VDP.

## Discussion

In this paper, we show zELK channels exhibit VDP that results from the channel undergoing a slow state-dependent transition to a mode with a more favorable opening transition. We then show that the VDP transition involves an interaction between the intracellular N-terminal eag domain and C-terminal CNBHD. Combining transition metal ion FRET, patch-clamp fluorometry, and incorporation of a fluorescent noncanonical amino acid, we show a rearrangement between the eag domain and CNBHD that exhibits the same ATP/Mg$^{2+}$-dependence, voltage-dependence, and kinetics as VDP. We proposed that this rearrangement of the eag domain-CNBHD interaction is coupled to channel opening and underlies VDP in these channels (*Figure 6D*). VDP of mammalian ELK and ERG channels appears to be an adaptation to dampen the hyperexcitability of neurons and cardiac tissue.

Previously, it has been shown that hELK1 is downregulated by PI(4,5)P$_2$ (*Li et al., 2015*). Similarly, we found that excision of the patch causes a run-up of the current that is prevented by the presence of ATP/Mg$^{2+}$. This downregulation by PI(4,5)P$_2$ is unusual in ion channels which are generally upregulated by PI(4,5)P$_2$ (*Hille et al., 2015*). Furthermore, for both hELK1 (*Li et al., 2015*) and zELK, PI(4,5)P$_2$ degradation leaves the channels in a potentiated mode that no longer undergoes VDP. These results suggest that VDP could result from a simple mechanism where PI(4,5)P$_2$ binds with higher affinity to the closed state of the channel than the open state, and unbinds slowly upon depolarization. This mechanism could account for our ATP/Mg$^{2+}$-dependence and voltage-dependence of VDP. It would suggest that PI(4,5)P$_2$ regulation is linked to a rearrangement of the eag domain-CNBHD interaction. However, hERG channels are also thought to undergo VDP (*Tan et al., 2012*; *Goodchild et al., 2015*) but are not appreciably regulated by PI(4,5)P$_2$ (*Kruse and Hille, 2013*). The precise role of PI(4,5)P$_2$ in VDP has yet to be fully understood.

The interaction between the eag domain and CNBHD has been well studied in ERG and EAG channels. Direct interaction between the eag domain and CNBHD in hERG channels has been demonstrated by multiple approaches including FRET (*Gianulis et al., 2013*; *Gustina and Trudeau, 2009*, *2013*). In hERG channels, the eag domain-CNBHD interaction is necessary to maintain the slow deactivation and normal inactivation of the channel (*Gianulis et al., 2013*; *Gustina and Trudeau, 2009*, *2013*). Furthermore, the eag domain-CNBHD interface of ERG channels is altered in some forms of long QT syndrome and schizophrenia (*Chen et al., 1999*; *Huffaker et al., 2009*). In EAG channels, direct interaction between the EAG domain and CNBHD was demonstrated using X-ray crystallography and cryo-EM (*Haitin et al., 2013*; *Whicher and MacKinnon, 2016*). Indeed, the eag domain-CNBHD complex has been shown to adopt two closely related but different conformations in X-ray crystallography (*Haitin et al., 2013*). These two conformations of the complex predict a small change in the distances between the eag domain and CNBHD. Breaking the critical salt bridge between eag domain and CNBHD significantly altered activation gating in EAG channels. Moreover, mutations at this interface of EAG channels have being associated with cancer (*Haitin et al., 2013*). For both ERG and EAG channels, the interaction has been shown to be intersubunit rather than intrasubunit (*Gianulis et al., 2013*; *Whicher and MacKinnon, 2016*). It appears that intersubunit eag domain-CNBHD interactions are a general self-regulatory mechanism among all three subfamilies of KCNH channels, EAG, ERG, and ELK; though the forms of regulation are different for the individual channels.

Our experiments with zELK suggest that the intracellular domains are responsible for the VDP. We hypothesize that, for KCNH channels, a rearrangement of the eag domain-CNBHD interaction is necessary for VDP. Moreover, we suggest that there is a rearrangement of the eag domain-CNBHD interaction coupled to the opening of the channel. Since the rearrangement is coupled to opening, it occurs preferentially at depolarized voltage and stabilizes channel opening. Indeed, thermodynamics dictates that any open state-dependent transition will stabilize channel opening. Therefore, while the molecular mechanism for VDP might be distinct for different channels, the presence of a slow state-dependent transition might be a general theme that underlies the VDP of all channels.

## Materials and methods

### Molecular biology

The full length *D. rerio* zELK construct (GI: 159570347) was synthesized (Bio Basic, Amherst, NY) and subcloned into a modified pcDNA3 vector that contained a C-terminal YFP, a T7 promoter and 3'

and 5' untranslated regions of a Xenopus $\beta$-globin gene. Point mutations were made using Quick-change II XL Site-Directed Mutagenesis kit (Agilent technologies, Santa Clara, CA). The chimeras and deletions were made using standard overlapping PCR followed by ligation using T4 ligase or Gibson Assembly (New England Biolabs). The sequences of the DNA constructs were confirmed by fluorescence-based DNA sequencing (Genewiz LLC, Seattle, WA). The RNA was synthesized in vitro using HiScribe T7 ARCA mRNA Kit (New England Biolabs, Ipswich, MA) or mMESSAGE mMACHINE T7 ULTRA Transcription Kit (ThermoFisher, Waltham, MA) from the linearized cDNA. mEAG1 was a gift from Dr. Gail Robertson (University of Wisconsin-Madison, Madison, WI).

## Heterologous expression and electrophysiology

Xenopus oocytes were prepared as previously described (*Varnum et al., 1995*). The pANAP plasmid cDNA (~50 nL of 100 ng/ml) containing the orthogonal tRNA/aminoacyl-tRNA synthetase specific to L-Anap (*Chatterjee et al., 2013*) was injected into the Xenopus oocyte nucleus. L-Anap (~50 nL of 1 mM free-acid form, AsisChem, Waltham, MA) as well as channel mRNA were injected into the cytosolic regions of oocytes separately. 2 to 4 days after injection, currents were recorded in the cell-attached and inside-out configuration of the patch-clamp technique using an EPC-10 (HEKA Elektronik, Germany) or Axopatch 200B (Axon Instruments, Union City, CA) patch-clamp amplifier and PATCHMASTER software (HEKA Elektronik). For oocyte patch-clamp recording, the standard bath and pipette saline solutions contained 130 mM KCl, 10 mM HEPES, 0.2 mM EDTA, pH 7.2. For patch-clamp fluorometry, 0.5 mM niflumic acid was added to the bath solution and the perfusion solution to remove calcium-activated $Cl^-$ currents. Different concentrations of $Co^{2+}$ were added to the perfusion solution with EDTA eliminated. Borosilicate patch electrodes were made using a P97 micropipette puller (Sutter Instrument, Novato, CA). The initial pipette resistance was 0.3–0.7 M$\Omega$ for oocyte recordings. Recordings were made at 22°C to 24°C.

The channel conductance-voltage relationship (G-V curve) was measured from the instantaneous tail currants at $-100$ mV as a function of the voltage of the main pulse. It was fitted with a Boltzmann equation:

$I = I_{min} + (I_{max} - I_{min}) / (1 + exp[(V_{1/2} - V)/V_s])$

where $I_{max}$ is the maximum tail current at $-100$ mV, $I_{min}$ is the minimum tail current after hyperpolarizing voltage steps, V is the membrane potential, $V_{1/2}$ is the potential for half-maximal activation, and $V_s$ is the slope factor.

The change in Gibbs free energy of channel activation was calculated according to the following equation: $\Delta G = RTV_{1/2}/V_s$, where R is the gas constant, and T is temperature in kelvin. The VDP due to a prepulse was calculated using the following equation: $\Delta\Delta G$ (prepulse) = $\Delta G$ (after prepulse) - $\Delta G$ (before prepulse).

## Microscopy and fluorometry

Patch-clamp fluorometry (PCF) was performed using a Nikon Eclipse TE2000-E microscope with a 60X water immersion objective (N.A.=1.2). Epifluorescent recording of L-Anap was performed with wide-field excitation using a Lambda LS Xenon Arc lamp (Sutter Instruments), as well as a 376/30 nm excitation filter and 485/40 nm emission filter. YFP was excited with a 490/10 nm excitation filter and 535/30 nm emission filter. Images were collected with a 50 or 100 ms exposure using an Evolve 512 EMCCD camera (Photometrics, Tucson, AZ) and MetaMorph software (Molecular Devices, Sunnyvale, CA). VC3-8xP series valve-controlled pressurized perfusion system (Scientific Instruments, Farmingdale, NY) was used to minimize electronic noise during PCF experiments. For spectral measurements, images were collected by a Cascade 512B intensified CCD camera (Roper Scientific, Tucson, AZ) attached to a spectrograph (Acton research, Acton, MA) on the output port of the microscope. Spectra were analyzed by measuring line-scans across the patch area. Spectra were background subtracted using a line-scan of the non-fluorescent region outside of patch.

## tmFRET efficiency calculation

The tmFRET efficiency measured by the decrease in donor fluorescence upon addition of the metal acceptor can be affected by nonspecific decreases in fluorescence that do not involve FRET with the metal bound to the dihistidine motif. The FRET efficiency can be corrected for these nonspecific decreases in donor fluorescence using the fluorescence decrease for channels without the

dihistidine. The precise form of the correction depends on the source for the nonspecific fluorescence decrease. If the decrease comes from a source that does not involve energy transfer, such as static quenching, bleaching, inner filter effect, or nonspecific loss of the fluorophore, then the FRET efficiency can be calculated using the following equation:

$$FRET_{eff} = 1 - \frac{F_{HH}}{F_{noHH}}$$
(1)

where $F_{HH}$ is the normalized fluorescence of channels with dihistidines and $F_{noHH}$ is the normalized fluorescence of channels without dihistidines. In each case, the $F$ values are the fluorescence measured at metal concentrations that saturate the binding sites normalized by the fluorescence in the absence of metal, e.g.

$$F_{HH} = \frac{fl\,(metal)}{fl\,(no\,metal)}$$
(2)

If the nonspecific decrease in fluorescence is due to collisional quenching or FRET to a different metal ion bound to an endogenous metal binding site, then the mechanism of quenching involves an additional pathway for relaxation of the fluorophore from its excited state. Consider the following scheme for a fluorophore relaxing from the excited state.

$$
\begin{array}{c}
photon \\
\uparrow k_{ph} \\
nonspecific\ quenching \overset{k_{no}}{\leftarrow} F^* \overset{k_{HH}}{\rightarrow} FRET\ to\ metal\ dihistidine
\end{array}
$$

where $F^*$ is the excited state of the fluorophore, $k_{ph}$ is the rate constant for emission of a photon by the excited-state fluorophore, $k_{HH}$ is the rate constant for energy transfer to the metal bound to the dihistidine, and $k_{no}$ is the sum of the rate constants for nonspecific sources of energy transfer. $FRET_{eff}$ in terms of the rate constants is given by:

$$FRET_{eff} = \frac{k_{HH}}{k_{ph} + k_{HH}}$$
(3)

Rearranging:

$$1 + \frac{k_{HH}}{k_{ph}} = \frac{1}{1 - FRET_{eff}}$$
(4)

Furthermore, $F_{noHH}$ in terms of the rate constants is given by:

$$F_{noHH} = \frac{k_{ph}}{k_{ph} + k_{no}}$$
(5)

Rearranging:

$$\frac{k_{no}}{k_{ph}} = \frac{1}{F_{noHH}} - 1$$
(6)

Finally, $F_{HH}$ in terms of the rate constants is given by:

$$F_{HH} = \frac{k_{ph}}{k_{ph} + k_{HH} + k_{no}}$$
(7)

Rearranging:

$$\frac{1}{F_{HH}} = 1 + \frac{k_{HH}}{k_{ph}} + \frac{k_{no}}{k_{ph}}$$
(8)

Substituting in *Equations 4 and 6* into *Equation 8* gives:

$$\frac{1}{F_{HH}} = \frac{1}{1 - FRET_{eff}} + \frac{1}{F_{noHH}} - 1 \tag{9}$$

And solving for $FRET_{eff}$ gives:

$$FRET_{eff} = 1 - \frac{1}{1 + \frac{1}{F_{HH}} - \frac{1}{F_{noHH}}} = \frac{F_{noHH} - F_{HH}}{F_{HH} * F_{noHH} + F_{noHH} - F_{HH}} \tag{10}$$

Both equations (*Equations 1 and 10*) produce similar values of $FRET_{eff}$. when the value of $F_{noHH}$ is near one (little decrease in donor fluorescence for channels without the dihistidine), as seen for the experiments in this paper. Since the small nonspecific fluorescence decrease in these experiments likely involved collisional quenching or FRET to a metal ion bound to an endogenous metal binding site, we used *Equation 10* to calculate the tmFRET efficiency.

The distance (R) between L-Anap and the metal ion was calculated using the Förster equation: $R = R_0 (1/ FRET_{eff} - 1)^{1/6}$, where $R_0$ is the Förster distance for FRET between L-Anap and $Co^{2+}$-dihistidine (12 Å) (*Aman et al., 2016*).

## Statistics

All data were analyzed using IgoPro (Wavemetrics, Lake Oswago, OR). Data parameters were expressed as mean ± SEM of n experiments. Statistical significance ($p < 0.05$) was determined by using Student's *t* test.

## Acknowledgements

We thank Drs. SE Gordon, TK Aman, GE Flynn, MC Trudeau, MD Varnum, ZM James, EG Evans, EN Senning, JLW Morgan, TI Brelidze, AE Carlson, MC Puljung and JR Bankston for advice and assistance; and X Opitz-Araya for technical support. This work was supported by the National Institute of Mental Health under Grant R01MH102378 (to WNZ) and by the National Eye Institute of the National Institutes of Health under Grant R01EY010329 (to WNZ).

## Additional information

### Funding

| Funder | Grant reference number | Author |
|---|---|---|
| National Institute of Mental Health | R01MH102378 | William N Zagotta |
| National Eye Institute | R01EY010329 | William N Zagotta |

The funders had no role in study design, data collection and interpretation, or the decision to submit the work for publication.

### Author contributions

GD, Conceptualization, Methodology, Acquisition of data, Analysis and interpretation of data, Writing—original draft, Writing—review and editing; WNZ, Conceptualization, Supervision, Funding acquisition, Methodology, Project administration, Writing—review and editing

### Author ORCIDs

William N Zagotta, http://orcid.org/0000-0002-7631-8168

### Ethics

Animal experimentation: This study was performed in strict accordance with the recommendations in the Guide for the Care and Use of Laboratory Animals of the National Institutes of Health. All of the animals were handled according to an approved institutional animal care and use committee (IACUC) protocol (#2689-01 ) of the University of Washington.

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
