## [Decision Letter]

Thank you for submitting your article "Molecular mechanism of voltage-dependent potentiation of KCNH potassium channels" for consideration by *eLife*. Your article has been favorably evaluated by Richard Aldrich (Senior Editor) and three reviewers, one of whom, Baron Chanda (Reviewer #1), is a member of our Board of Reviewing Editors. The following individual involved in review of your submission has agreed to reveal their identity: Matthew Trudeau (Reviewer #3).

The reviewers have discussed the reviews with one another and the Reviewing Editor has drafted this decision to help you prepare a revised submission.

In this study, Dai and Zagotta explore the molecular and structural mechanisms that underlie voltage-dependent potentiation (VDP) in KCNH family of potassium channels. Previous studies have shown that VDP is eliminated in excised patches due to hydrolysis of PIP2. Here, the authors find that this potentiation can be maintained if supplemented by ATP and Mg. This ability to study VDP in excised patches allowed the authors to apply techniques such as patch clamp fluorometry to track conformational changes that underlie voltage-dependent potentiation. They find that the transition metal ion-FRET between labeled EAG domain and CNBHD increases upon depolarization and this increase is not observed in absence of ATP. They also show that VDP closely tracks the strength of interaction between these two domains; when an intersubunit salt bridge between EAG and CNBHD is disrupted, VDP is attenuated but it can be restored by charge swapping mutations.

This is a technical tour de force using a fluorescent non-canonical amino acid (L-ANAP) incorporation, tmFRET with ANAP-containing channels and Co^2+^, and patch-clamp fluorometry to simultaneously detect structural rearrangements and function in ELK channels. Overall, this is an elegant mechanistic study that illustrates at a molecular level how KCNH family of ion channels is potentiated by depolarizing voltage pulses. In addition, this study shines new light on why many disease causing mutations occur at the interface of these two domains which presumably disrupts VDP.

Essential revisions:

1) Figure 5: Even in absence of di-Histidines, ELK channels show some non-specific binding at that site. If it is non-specific binding at that site, why does it not change in a prepulse dependent manner (Figure 5)? How general are these non-specific effects? Were they observed in other TM-FRET experiments?

2) Figure 2. There is no quantification of the change in Tau in MgATP condition, as in 2E for the cell attached and no MgATP conditions.

3) Figure 3. What is the slopes of the G(V)s in the different conditions? If they are very different from each other’s, then ΔG would be a better comparison to use.

4) The FRET efficiency equation is not standard and needs to be better explained. In addition, it doesn't seem to have the right units (should be unit less, but now there are a product of fluorescence signal in the denominator and only single F values in the nominator).

5) My major comment is that a mechanism for the PIP2 (ATP/Mg^2+^) regulation and its relationship to voltage would be welcome in the kinetic scheme or cartoon scheme in order to guide future experiments.

---

## [Author Response]

*Essential revisions:*

*1) Figure 5: Even in absence of di-Histidines, ELK channels show some non-specific binding at that site. If it is non-specific binding at that site, why does it not change in a prepulse dependent manner (Figure 5)? How general are these non-specific effects? Were they observed in other TM-FRET experiments?*

The small quenching in the absence of dihistidine is not non-specific binding to that same site (which is no longer a possible binding site for Co^2+^), but is instead probably a mixture of collisional quenching, and Co^2+^ binding to another site. Therefore this background quenching did not change in a prepulse-dependent manner. We have clarified this in the text. These non-specific effects are common and have been observed in other tmFRET papers (e.g. Taraska et al., 2009). We have also inserted a section describing our FRET efficiency calculation and correction of the non-specific quenching in the Materials and methods.

*2) Figure 2. There is no quantification of the change in Tau in MgATP condition, as in 2E for the cell attached and no MgATP conditions.*

As suggested we have added the quantification in the ATP/Mg^2+^ condition to Figure 2. The data with ATP/Mg^2+^ in the inside-out configuration overlap nicely the data in the cell-attached configuration.

*3) Figure 3. What is the slopes of the G(V)s in the different conditions? If they are very different from each other’s, then DeltaG would be a better comparison to use.*

Good idea! We have included the ΔΔG values even though the slopes are not very different We calculated the ΔG for the channel activation and ΔΔG due to prepulses and updated Figure 3. We also added the equation for calculating ΔΔG in Materials and methods.

*4) The FRET efficiency equation is not standard and needs to be better explained. In addition, it doesn't seem to have the right units (should be unit less, but now there are a product of fluorescence signal in the denominator and only single F values in the nominator).*

We have added a section deriving the equation for the FRET efficiency in the Materials and methods. As indicated in that section, F_HH_ and F_no HH_ are normalized fluorescence and thereby unitless.

*5) My major comment is that a mechanism for the PIP2 (ATP/Mg^2+^) regulation and its relationship to voltage would be welcome in the kinetic scheme or cartoon scheme in order to guide future experiments.*

We have added PIP_2_ to Figure 6 to guide future experiments. We have also added a sentence in the figure legend speculating the mechanism of the PIP2 regulation.